# Higher Seasonal Variation of Systemic Inflammation in Bipolar Disorder

**DOI:** 10.3390/ijms25084310

**Published:** 2024-04-13

**Authors:** Sara Dallaspezia, Vincenzo Cardaci, Mario Gennaro Mazza, Rebecca De Lorenzo, Patrizia Rovere Querini, Cristina Colombo, Francesco Benedetti

**Affiliations:** 1Division of Neuroscience, IRCCS Ospedale San Raffaele, 20132 Milano, Italy; dallaspezia.sara@hsr.it (S.D.); mazza.mariogennaro@hsr.it (M.G.M.); 2Università Vita-Salute San Raffaele, 20132 Milano, Italy; cardaci.vincenzo@hsr.it (V.C.); rovere.patrizia@hsr.it (P.R.Q.); colombo.cristina@hsr.it (C.C.); 3Division of Immunology, Transplantation and Infectious Diseases, IRCCS Ospedale San Raffaele, 20132 Milano, Italy; delorenzo.rebecca@hsr.it; 4Mood Disorders Unit, IRCCS Ospedale San Raffaele, 20132 Milano, Italy

**Keywords:** bipolar disorder, obsessive compulsive disorder, season, inflammation, neutrophil, lymphocyte, platelet, serotonin, melatonin

## Abstract

Seasonal rhythms affect the immune system. Evidence supports the involvement of immuno-inflammatory mechanisms in bipolar disorder (BD), with the neutrophil to lymphocyte ratio (NLR), and the systemic immune-inflammatory index (SII; platelets × neutrophils/lymphocytes) consistently reported to be higher in patients with BD than in HC, but seasonal rhythms of innate and adaptive immunity have never been studied. We retrospectively studied NLR and SII in 824 participants divided into three groups: 321 consecutively admitted inpatients affected by a major depressive episode in course of BD, and 255 consecutively admitted inpatients affected by obsessive–compulsive disorder (OCD; positive psychiatric control), and 248 healthy controls (HC). Patients with BD showed markedly higher markers of systemic inflammation in autumn and winter, but not in spring and summer, in respect to both HC and patients with OCD, thus suggesting a specific effect of season on inflammatory markers in BD, independent of a shared hospital setting and drug treatment. Given that systemic inflammation is emerging as a new marker and as target for treatment in depressive disorders, we suggest that seasonal rhythms should be considered for tailoring antidepressant immuno-modulatory treatments in a precision medicine approach.

## 1. Introduction

Clinical epidemiology affirms clear seasonal effects in the pattern of recurrence of severe illness episodes and hospitalization in bipolar disorder (BD), with excess depressive and mixed episodes in autumn and spring, and mania in summer [1]. Changes in mood across seasons are parallel to the effects of changes in the photoperiod on brain neurotransmitters, with serotonin (5-HT) release increasing, and the serotonin transporter density decreasing, together with the amount of daily light [2,3], a correlational effect enhanced in patients with mood disorders and associated with the severity of depression in winter, when 5-HT is lower and rapidly cleared from the extracellular space by more transporters [4,5].

In recent years, an unprecedented amount of evidence has supported immuno-inflammatory mechanisms as a point of convergence of several pathways implicated in BD, with systemic low-grade inflammation, T cells’ senescence and increased innate immunity, raised pro-inflammatory setpoints, increased the production of cytokines, activated gene expression in circulating monocytes, activated brain microglia, and prompted secondary disturbances of neurotransmitter systems [6,7,8]. The interpretation of these findings is still under debate, and several different mechanisms have been proposed, including an inborn dysregulation of the immune system, leading to auto-inflammatory reactivity, stress, and exposure to infectious agents [9,10,11,12,13].

The clinical relevance of raised immune-inflammatory setpoints as reliable biomarkers for the definition of new endophenotypes in mood disorders consistently emerges from the recent literature [14]. Peripheral levels of inflammatory cytokines and circulating cytotoxic cell counts are associated with the disruption of white matter integrity and choroid plexus hypertrophy [15,16], predicting worse cognitive functions and worse antidepressant response in BD [17,18,19]. The neutrophil to lymphocyte ratio (NLR), and the systemic immune-inflammatory index (SII; platelets x neutrophils/lymphocytes), easily calculated from white blood cell assays, are higher in patients with mood disorders, and associated with lower hippocampal volumes, and lower responses to antidepressant treatment [20,21]. In patients with higher baseline inflammatory biomarkers, new treatments affecting the immune system (including infliximab, minocycline, celecoxib, low-dose interleukin 2) can boost antidepressant response in patients not responding to conventional monoaminergic antidepressants [22,23,24,25], possibly depending on the basal levels of inflammation or on other not-yet-studied predictive factors [23,26,27], and thus raising the possibility of individually tailored immune modulation in patients with treatment-resistant depression (TRD) [28,29].

Unfavorable outcomes of available antidepressant treatments is an issue of particular clinical relevance in patients with BD, who show higher degrees of antidepressant drug treatment failures in clinical settings [30,31,32]. Possibly, as a consequence of their debilitating condition, about 30% of patients with BD attempt suicide [33,34], and about 20% eventually die from suicide [35,36], and which is associated with higher density and pro-inflammatory activation status of brain microglia [37,38], and follows seasonal rhythms [39]. Considering that TRD is currently estimated to develop in 30% of patients with major depression [40], the precise individual definition of inflammatory biomarkers which are clinically relevant is now a priority in mood disorders research.

Seasonal changes in the photoperiod affect the immune system, with lymphocyte-mediated adaptive immunity being depressed, and innate immunity more activated, in winter [41], an effect at least partially mediated by seasonal changes in brain function through the secretion of melatonin, which activates monocytes [42] and neutrophils [43], and exerts complex effects on lymphocyte proliferation and activation status [41]. These seasonal changes in immune function have been recently confirmed in a large human population study [44]. A single, pivotal study showed that, in respect to healthy controls, a group of 20 patients with major depressive disorder with a seasonal winter pattern showed higher macrophage activity and production of cytokines, and lower lymphocyte proliferation in winter, a difference normalized by treatment with bright-light therapy [45], which can also normalize 5-HT function in winter depression [46]. This provocative observation supported the association between the seasonal availability of light, according to the changing photoperiod, and the immune function in patients with mood disorders.

Notwithstanding the above, differences in seasonal changes of immune-inflammatory biomarkers in patients with mood disorders have not yet been studied. A consistent section of the literature affirms that neutrophils tend to increase in autumn/winter, but NLR is stable in healthy adults across seasons, while it shows some seasonal variation in children [47]. On the other hand, there is sparse evidence that NLR can show seasonal variation (higher in autumn/winter) in some medical conditions associated with systemic inflammation [48].

Here, we hypothesized that patients with BD would experience higher seasonal changes in immune-inflammatory circulating cell counts, as summarized by the NLR and SII index. We tested this hypothesis by comparing blood cell counts in respect both to healthy participants working in the hospital and to patients with another psychiatric condition, such as obsessive–compulsive disorder (OCD), thus aiming at controlling the effects of interest for the exposure of the participants to psychotropic drug treatments and to the hospital environment. The possible seasonal variation of administered drug classes and of body mass index was also evaluated.

## 2. Results

The demographic characteristics and blood cell counts of participants, divided according to diagnostic groups and to season of hospitalization, are presented in Table 1. The age and sex distributions of the participants were not significantly different among seasons.

Inspection of data (Figure 1) shows that patterns of change of NLR and SII across seasons did not follow parallel slopes of time course in the diagnostic groups. A GLZM separate slopes regression confirmed the statistical significance of this effect for both NLR (main effect of diagnosis: χ^2^ = 6.210, *p* = 0.0448; diagnosis x season interaction: χ^2^ = 21.348, *p* = 0.0455) and SII (main effect of diagnosis: χ^2^ = 7.159, *p* = 0.0279; diagnosis × season interaction: χ^2^ = 42.206, *p* = 0.0122).

This was due to higher inflammatory biomarkers in patients with BD in autumn and in winter, with significant differences for SII among diagnostic groups in autumn (GLZM homogeneity of slopes regression, main effect of diagnosis: (χ^2^ = 6.728, *p* = 0.0346) and in winter (χ^2^ = 18.935, *p* = 0.00008), and for NLR in autumn (χ^2^ = 14.807, *p* = 0.0006), with trend differences in winter (χ^2^ = 5.202, *p* = 0.074).

The effect size of the observed group differences was small in autumn (SII: _p_η^2^ = 0.030; NLR: _p_η^2^ = 0.023), leading to a low achieved power of the sample to detect significance (SII: λ = 6.74, power 0.63; NLR: λ = 5.19, power 0.51); the effect size was medium in winter (SII: _p_η^2^ = 0.082; NLR: _p_η^2^ = 0.065), leading to a large power to detect significance (SII: λ = 19.50, power 0.98; NLR: λ = 15.11, power 0.94).

Differences in NLR and SII between patients with OCD and HC were not significant at any time point.

BMI was associated with blood cell counts, but not with the immune-inflammatory measures NLR and SII. In the whole sample, BMI positively correlated with neutrophils (r = 0.219, *p* < 0.001) and lymphocytes (r = 0.156 *p* < 0.001) counts, but not with platelet counts (r = 0.103, *p* = 0.150). It did not correlate neither with NLR (r = 0.013, *p* = 0.720), nor with SII (r = 0.034, *p* = 0.360). This profile was quite homogeneous among diagnostic subgroups, except for lymphocytes counts.

(i) In HC, BMI positively correlated with neutrophils (r = 0.230, *p* < 0.001) and lymphocytes (r = 0.185, *p* = 0.004) counts, but not with platelet counts (r = 0.0.61, *p* = 0.345). It did not correlate neither with NLR (r = 0.038, *p* = 0.564), nor with SII (r = 0.064, *p* = 0.327). (ii) In patients with BD, BMI positively correlated with neutrophils counts (r = 0.181, *p* = 0.001), but not with lymphocytes (r = 0.089, *p* = 0.118) and platelet counts (r = −0.030, *p* = 0.603). It did not correlate neither with NLR (r = −0.006, *p* = 0.914), nor with SII (r = 0.008, *p* = 0.896). (iii) In patients with OCD, BMI positively correlated with neutrophils (r = 0.175, *p* = 0.014) and lymphocytes (r = 0.189, *p* = 0.008) counts, but not with platelet counts (r = 0.103, *p* = 0.150). Again, it did not correlate neither with NLR (r = −0.042, *p* = 0.561), nor with SII (r = 0.013, *p* = 0.851). The difference in the relationship of BMI with lymphocyte counts in BD was significant, as confirmed using a GLZM separate slopes regression, showing a significant BMI × diagnosis interaction on lymphocyte counts (LRχ^2^ = 23.60 *p* < 0.00001).

A GLZM analysis of covariance with BMI as the dependent variable and season, with age and sex as factors, showed (i) a significant effect of age in all diagnostic groups (HC: LRχ^2^ = 7.19, *p* = 0.008; BD: LRχ^2^ = 4.75, *p* = 0.029; OCD: LRχ^2^ = 8.73, *p* = 0.003), with older age being associated with higher BMI (Spearman r = 0.172, *p* < 0.001); (ii) a significant effect of sex in HC and OCD, but not in BD (HC: LRχ^2^ = 20.97, *p* < 0.001; OCD: LRχ^2^ = 7.76, *p* = 0.005), with higher BMI in males (Mann–Whitney U Test, Z = 5.848, *p* < 0.001); and (iii) a significant effect of season only in BD (LRχ^2^ = 9.37, *p* = 0.025), with lower BMI in autumn than in spring and in winter (Kruskal–Wallis test: H = 11.11, *p* = 0.0111; post hoc multiple comparison: autumn vs. spring H = 2.72, *p* = 0.039; autumn vs. winter H = 2.84, *p* = 0.0268).

Given the observation of seasonal changes of BMI in BD, we also tested if the relationship between BMI and immune-inflammatory indexes could differ among seasons, but a GLZM separate-slopes regression with age, sex, season, and BMI as factors, yielded non-significant interactions both for NLR and for SII.

The season did not significantly influence ongoing drug treatments at the time of blood sampling (Table 2), while diagnosis was associated with a higher rate of antidepressant drugs administration in OCD than in BD, and a higher rate of anticonvulsant drugs administration in BD than in OCD. Lithium was being administered to patients with BD only, and the rate of antipsychotic drugs administration did not significantly differ among the two groups.

## 3. Discussion

This is the first observation that patients with bipolar disorder show higher markers of systemic inflammation in autumn and winter in respect to both HC and patients with OCD, thus suggesting a specific effect of season on inflammatory markers in BD.

This finding is in agreement with a single previous study on winter depression in MDD, reporting lower lymphocyte proliferation in patients [45], thus supporting the hypothesis of different seasonal changes of inflammatory markers in patients with mood disorders in respect to patients affected by other psychiatric diagnoses and to healthy conditions. These effects are unlikely to be explained by ongoing drug treatments, which did not differ across seasons, and by BMI, which showed a seasonal variation in patients with BD, but was unrelated with systemic inflammatory measures in all groups and in all seasons. Moreover, we replicated the existing findings of a lack of seasonal variation of NLR in HC, supporting the hypothesis of a specific effect in patients with BD, and we first reported the lack of seasonal variation of immune-inflammatory markers in OCD. Again, this suggests that seasonal variation in inflammation could be a characteristic of mood disorders, because OCD has been associated both, with signs of brain activation of microglia [49], with possible positive effects of adjunctive anti-inflammatory medication [50], and with proposed possible seasonal variation in symptoms, albeit disconfirmed in most reports [51,52,53,54,55,56]. Further studies are needed to explore this possible specificity, taking into account the psychiatric conditions which have been associated with altered immuno-inflammatory setpoints.

Our present observation could contribute to crucial issues for clinics and research in mood disorders.

First, there is now clear evidence of a relationship between inflammation and brain structure and function in mood disorders. Activation of innate immunity has a major impact on brain structure and function in BD, with levels of peripheral cytokines and immune cell composition being associated with the white matter microstructure, brain metabolites, effect of antidepressant treatments, and cognitive impairment [16,57,58,59]. This also involves a decrease in 5-HT function mediated by tryptophan breakdown and by activation of the kynurenine pathway [7,60,61], which in turn impacts immune functions [62]. Interventions targeting the immune system could counteract these effects of altered inflammatory setpoints on the brain homeostasis of patients with mood disorders. In this respect, it should be noted that a consistent area of the literature reported higher immune inflammatory setpoints in psychiatric conditions, including mood disorders, schizophrenia, anxiety disorders, post-traumatic stress disorder, autism spectrum disorders, sleep disorders, and OCD [63,64,65,66,67]. However, available studies did not consider the seasonal variation in immune markers. This is clinically relevant in light of the ongoing effort in defining thresholds for these measures, potentially affecting treatment course and the outcome of the disorder. We suggest that future studies should take into account this variable, in order to provide unbiased estimates of the immunological phenotype associated with psychiatric conditions, which could vary across seasons.

Second, patients affected by mood disorders seem to be more vulnerable to the detrimental effects of infections by pathogens, as suggested by reports of a double rate of mortality for COVID-19 [68], of the relationship between infections and the subsequent development of new and recurrent major depressive episodes [69,70], and of atypical immune responses to pathogens [71]. It can be surmised that different immune-inflammatory setpoints during the cold season could foster, and/or be a consequence of the above, and mark a particular sensitivity to the detrimental effects of pathogens, and should be managed in clinics.

Third, it is surmised that the seasonality of immune-inflammatory function could contribute to explaining discrepancies in the effects of antidepressant treatments targeting both monoaminergic neurotransmitters and the immune system (see Introduction). A seasonal variation in antidepressant prescriptions in population-wide samples, with a peak in autumn and early winter, has been consistently reported [72,73], and seasonal pharmacotherapy has been proposed for major depression with typical seasonal winter patterns of recurrence [74]. In animal models, powerful drugs enhancing monoaminergic neurotransmission, such as imipramine, desipramine, amitriptyline, amineptine, mianserin, showed a seasonal variation in antidepressant-like behavioral responses, with effects peaking in March [75]. Possible seasonal effects of antidepressants in humans are suggested by the pivotal observations of higher rates of suicide when antidepressants are started in summer and autumn [76], and of a faster response to paroxetine when the photoperiod is increasing, from February to April, in respect to autumn [77]. Higher inflammation associated with major depression in the cold months could underpin a lower efficacy of conventional antidepressants, and be the target for specific immuno-modulatory interventions.

Finally, a major depressive episode in course of BD is a difficult-to-treat condition, with extremely low success rates of antidepressant drugs in naturalistic settings [31]. The use of monoaminergic antidepressants, which showed high efficacy in major depressive disorder, has been criticized in BD [32]. However, despite mood stabilizers being the mainstay for the treatment of depression in BD, the British Association for Psychopharmacology [78], the French Society for Biological Psychiatry and Neuropsychopharmacology [79], the American Psychiatric Association [80], the Canadian Network for Mood and Anxiety Treatments (CANMAT), and the International Society for Bipolar Disorders [81] still consider the combination of antidepressants and lithium as a possible option for this difficult condition, even as second-line treatment in respect to mood stabilizers alone. The main indication for antidepressants in bipolar depression comes from extrapolation of the strong unipolar data, given the absence of proven differences in the underlying biology of bipolar and unipolar depressed states [82]. Nevertheless, patients with BD spend a substantial proportion of their time ill [83], with depression representing their predominant abnormal mood state [84], facing poor prospective responses to antidepressant drug treatment [30].

On the other hand, neurobiological models of behavior allowed the development of nonpharmacological biological chronotherapeutic antidepressant treatments based on controlled exposure to environmental stimuli that act on biological rhythms, to which patients with bipolar depression are extremely sensitive [85]. Manipulations of the sleep–wake cycle such as sleep deprivation and sleep phase advance, and controlled exposure to bright light in the morning (light therapy), provide chronotherapeutic interventions, which can prompt the rapid remission of depressive symptoms [86,87,88], including life-threating behaviors such as suicidal ideation and planning [89].

The circadian and the immune systems closely interact. The master pacemaker in the suprachiasmatic nucleus (SCN) in the hypothalamus regulates neuronal activity, body temperature and hormonal signals, with translational–transcriptional feedback loops at the gene level, which are synchronized by environmental stimuli mainly involving exposure to light and darkness. The clock molecular machinery is expressed in peripheral tissues, where timing is synchronized centrally to ensure homeostasis. Disruption of the circadian timing system can then pave the way to several medical and psychiatric conditions across a lifespan, and is majorly correlated with all mood disorders, including BD [90,91].

The molecular clock gene machinery is expressed in immune cells, which show a daily fluctuation in activity, production of cytokines, development, and tissue trafficking in homeostatic or inflammatory conditions [92]. In homeostatic conditions, the immune system is actually under circadian control, with leukocyte counts peaking in circulating blood during the resting phase of the organism, and recruitment to tissues mainly occurring during the active phase, and with circadian timing of activity that responds to pathogens and promotes tissue recovery and the clear harmful elements from the circulation [93]. In turn, immune cells contribute to maintaining circadian homeostasis by directly affecting the molecular clock when infiltrating peripheral tissues [94]. It is now recognized that brain–immune interactions involve neural control of immune cell trafficking, with neural regulation of cell egress from bone marrow and tissue infiltration which is achieved via hormone and neurotransmitter release [95].

The circadian control of the immune system and the relationship between raised immuno-inflammatory setpoints and biological rhythms disruption has yet not been properly studied in mood disorders [96], but preliminary evidence in a small sample suggests a disruption in the circadian production of cytokines, associating it with depressive psychopathology [97], including daily behavioral rhythms disruption [98].

Both light therapy alone, and the combination of sleep deprivation and light, can reduce the levels of peripheral markers of inflammation in patients responding to treatment, possibly favoring adaptive over innate immunity mechanisms [45,99]. These treatments directly target the biological clock in the suprachiasmatic nucleus, and in humans they affect the circadian genes’ transcriptomics, with gene set analyses highlighting strong effects on pathways involved in immune function and inflammatory responses [100]. Given that poor response to antidepressant drugs is associated with raised inflammatory markers [19,101], including NLR [21], a higher activation of inflammation in patients with BD could contribute to explaining their poor response to drugs, and specific sensitivity to chronotherapeutics, in the cold seasons [102].

All these issues are open for further research, which may also address the main limitation of the present study, which is its cross-sectional design, with one observation for each participant. Prospective studies, assessing time-lagged variation in inflammatory markers across seasons in the same group with multiple sampling, are needed to clarify the specific patterns of change associated with these psychiatric conditions. Moreover, studies on the seasonal variation of other immune-inflammatory measures, such as CRP and cytokines, will help us to be precise about the possible specific immune mechanism underpinning the observed seasonal variation in NLR and SII.

Strengths of the present study include a focused research question and state-of-the-art laboratory methods, but our results must be viewed in light of some limitations. No patient was drug-naive, and the drug treatments administered during the course of the illnesses, including antidepressant and antipsychotic drugs, could have influenced laboratory measures. The lack of a control for possible specific effects of drugs on immunity limits the interpretation of the findings; however, the lack of a seasonal difference in ongoing drug treatments makes it unlikely that the observed seasonal differences in immune-inflammatory measures could be explained by drug effects. Generalizability to the general population and to physiological mechanisms, with effects which are expected to be different from patients with mood disorders, is not possible. Also, results could be affected by lifestyle (e.g., sleep and nutrition) and genetic factors’ effects, data not collected in our samples. Recruitment took place in a single center and in a single ethnic group, thus raising the possibility of population stratifications. The effect size was medium for the observed effects in autumn, suggesting a limited power to detect the significance of the observed differences, while it was medium in winter, suggesting a greater power. Future studies are expected to define the most appropriate sample size, by exploiting the observations of our exploratory research.

These limitations, however, do not bias the main finding of an increased seasonal variation in circulating white cells counts in patients with BD, associated with the autumn–winter seasons. In conclusion, the research perspective which lead to the present observation may help to define new biomarkers, and possible targets for treatment, to address an unmet need of precision medicine and foster personalized monitoring and treatment of this highly debilitating psychiatric condition [14], by considering seasonal rhythms when assessing immuno-inflammatory setpoints in patients with mood disorders and when defining strategies for pharmacological and non-pharmacological antidepressant interventions, and when investigating the effects of new antidepressant immuno-modulatory treatments.

## 4. Materials and Methods

We retrospectively studied 824 participants divided into three groups: 321 consecutively admitted inpatients affected by a major depressive episode in course of BD; 255 consecutively admitted inpatients affected by obsessive-compulsive disorder (OCD); and 248 healthy controls (HC).

The BD group included patients free of inflammation-related symptoms, including fever and infectious or inflammatory disease; uncontrolled systemic disease; uncontrolled metabolic disease or another significant uncontrolled somatic disorder; and somatic medications known to affect the immune system, such as corticosteroids, non-steroid anti-inflammatory drugs, and statins.

The HC group included people matched for age and sex, working in the same hospital where the patients had been hospitalized, and meeting “super-healthy” criteria, by excluding the presence of acute infectious diseases, hepatic liver or kidney disease, congestive heart failure, known malignancy, neurological disorders, pregnancy, taking medications for dyslipidemia, hyperglycemia, hypertension, insulin resistance, or obesity.

The OCD group met the same exclusion criteria of BD, and served as a positive control for (i) having a psychiatric condition; (ii) being hospitalized in the same hospital at the time of blood sampling; and (iii) being administered psychotropic treatments for a lifetime, often including a similar choice of drugs [103]. This design ensured a control condition for the possible effects of drugs on the immune system [28], and for exposure to pathogens and other environmental variables in hospital. None of the patients with OCD had a major depressive episode when tested of inflammatory markers.

Additional inclusion criteria for patients were absence of other psychiatric diagnoses; absence of pregnancy; history of epilepsy; major medical and neurological disorders; no treatment with long-acting neuroleptic drugs in the last three months before admission; absence of a history of drug or alcohol dependency within the last six months.

Patients had been consecutively admitted to the Mood Disorder unit of hospital ‘San Raffaele Turro’ in Milano, part of the Scientific Institute Ospedale San Raffaele, a university tertiary care hospital directly funded by the Italian Ministry of Health for the research and treatment of medical conditions, including psychiatric diseases. Patients had been referred by their general practitioner, or by the psychiatrist in charge of their outpatient treatment for a psychopathological condition needing hospitalization for treatment. Ongoing treatment(s) at admission were being administered based on clinical needs. No patient was in remission. Referral diagnosis was confirmed by the senior psychiatrist in charge of the patient for hospital treatment, under the supervision of the head of the ward (CC). After evaluation and diagnosis by the team responsible for the admission to the ward (coordinated by a vice-head of the ward), a senior psychiatrist experienced in mood disorders confirmed the diagnosis following a best estimate procedure, by interviewing subjects, family members, previous health professionals, and obtaining records when possible. The agreement of three senior independent raters was then required to define the diagnosis, and the lack of comorbid psychiatric and medical conditions according to the inclusion criteria of the study. Patients were included if the diagnosis was confirmed at discharge. Sociodemographic and clinical data were collected using a data extraction form.

After complete description of the study to the subjects, written informed consent was obtained. All the research activities were carried out according to the guidelines of the Declaration of Helsinki, and approved by the Ethics Committee of Ospedale San Raffaele in the framework of an ongoing prospective observational study about biomarkers and outcomes of psychiatric conditions (Protocol 10-06 SO, 30 June 2006). From among the studied population, matched groups of participants with BD and OCD were selected based on inclusion criteria for the specific purpose of this study.

Fasting blood sampling was performed in the context of routine blood examinations, in the morning, on the second day of hospitalization (in patients), or during routine preventive medicine checks by hospital personnel who served as HC. Blood was collected after overnight fasting (9–12 h) from the antecubital vein into EDTA anticoagulant using vacuum tubes (BD Biosciences, 368,856 vacutainers, Becton Dickinson Italia, Milano, Italy). Cell blood counts were obtained using the Sistema Sysmex Serie XN 9000 (Sysmex Europe, Norderstedt, Germany) automated hematology analyzer. Data on absolute cell counts were extracted from medical charts. One single time point was available for each participant.

As measures of systemic inflammation, we calculated the NLR, and the SII, which have been reportedly higher in patients with BD than in HC [104,105], and normal in OCD [106]. These measures are low-cost and reproducible tests easily calculated from white blood cell assays that can be determined under simple laboratory conditions, and they reflect the balance between innate and adaptive immunity in medical and psychiatric conditions [69,104]. Provided that our study focused on the effects of low-grade sub-clinical inflammation, to exclude the possible confounding effects of undiagnosed infective processes or other pathological conditions, only patients with neutrophil and lymphocyte counts within normal ranges (1500 to 7700 cells/µL for neutrophils, 1000 to 4800 cells/µL for lymphocytes) were included in the analysis.

Seasons were defined based on summer and winter solstices, and spring and autumn equinoxes, which mark the direction of daily change of the photoperiod (increasing during winter and spring, decreasing during summer and autumn): spring from 21 March to 21 June; summer from 21 June to 23 September; autumn from 23 September to 21 December; and winter from 21 December to 21 March.

To test the effects of season on inflammatory markers, and considering the a priori expected interaction of categorical and continuous independent factors (season, diagnosis, age, sex) and the non-normal distribution of inflammatory biomarkers, variables were entered into a generalized linear model (GLZM) analysis of homogeneity of slopes or separate-slopes regression as appropriate, with an identity link function [107]. Parameter estimates were obtained with iterative re-weighted least squares maximum likelihood procedures. The significance of the effects was calculated with the likelihood ratio (LR) statistic, which provides the most asymptotically efficient test known. By performing sequential tests for the effects in the model of each of the factors of the dependent variable, at each step adding an additional effect into the model, contributes to incremental Chi-square statistics, thus providing a test of the increment in the log-likelihood that is attributable to each currently estimated effect.

To provide the readers with enough information to assess the magnitude of the observed effect, we also calculated the effect size of the difference (partial η^2^), which is the proportion of the variability in the dependent variable(s) that is explained by the effect(s) of interest [108,109], following standard interpretation guidelines (_p_η^2^ ≥ 0.01 for a small effect; _p_η^2^ ≥ 0.06 for a medium effect; _p_η^2^ ≥ 0.14 for a large effect) [110].

The sample size included patients hospitalized, in this framework, between 2010 and 2020. We excluded patients hospitalized after the start of the COVID-19 pandemic for the possible interference of SARS-CoV-2 infection on immune-inflammatory measures. The exploratory nature of the study (which is the first, addressing the issue of possible seasonal variation in immune-inflammatory markers in BD) made an a priori estimation of the expected effect size that was not possible, and an a priori power calculation was then not applicable. The achieved power was calculated based on the observed effect size, at two-tailed α = 0.05.

All the statistical analyses were performed with a commercially available software package (StatSoft Statistica 12, Tulsa, OK, USA) and following standard computational procedures. All analyses were corrected for age and sex.

Given the reported association of body mass index (BMI) with both, immune-inflammatory markers in psychiatric conditions [111,112], and with the detrimental outcomes of BD, such as brain volume and microstructure abnormalities [113,114,115], and lower efficacy of antidepressant interventions [57], we also tested for seasonal variation in BMIs in the diagnostic groups, and for their possible association with NLR and SII. Moreover, we tested for group differences in ongoing drug treatments, at the moment of blood sampling, and for seasonal variation in drugs administered to participants.

## Figures and Tables

**Figure 1 ijms-25-04310-f001:**
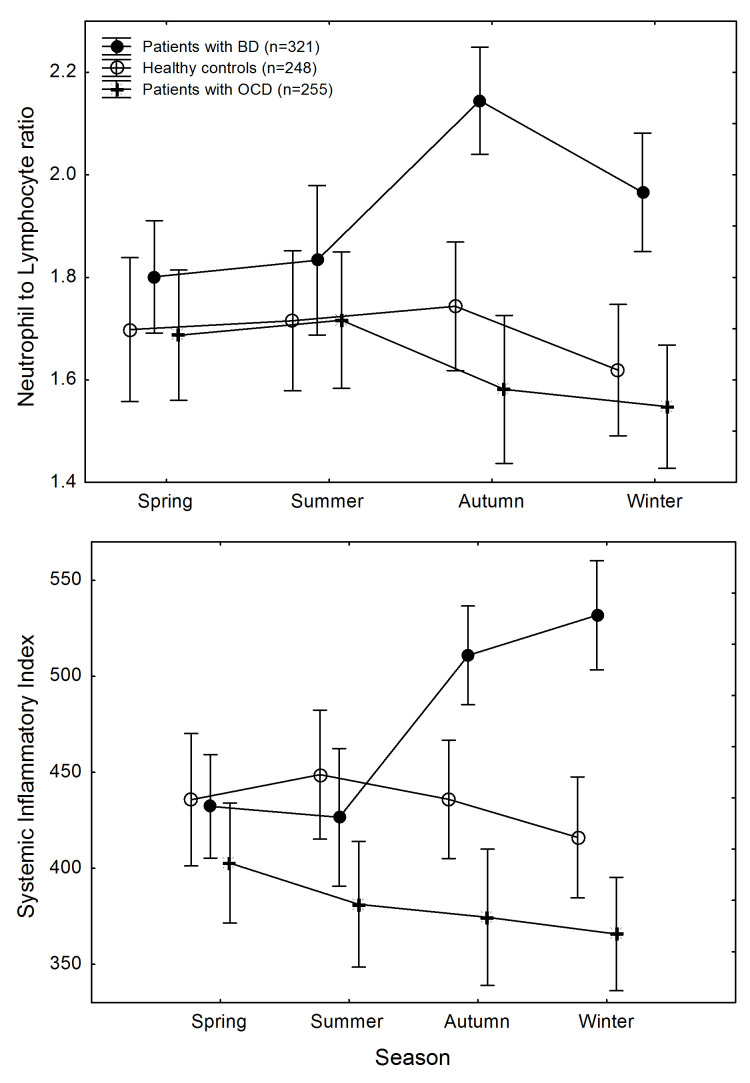
Changes in systemic inflammation across seasons in participants, divided according to diagnostic groups. **Top**: neutrophil to lymphocyte ratio. **Bottom**: systemic immune-inflammatory index. Points are means; whiskers are standard errors of the mean.

**Table 1 ijms-25-04310-t001:** Demographic characteristics of the participants divided according to diagnosis and to season, and their mean + SD blood cell counts, systemic inflammation indexes, and BMIs.

	Healthy Controls (*n* = 248)			
	Spring (*n* = 55)	Summer (*n* = 58)	Autumn (*n* = 69)	Winter (*n* = 66)
Sex (M/F)	23/32	19/39	28/41	24/42
Age	43.8 ± 12.3	46.3 ± 10.5	45.3 ± 11.3	44.2 ± 9.9
BMI	24.12 ± 3.83	24.73 ± 3.90	23.83 ± 3.96	23.88 ± 3.52
Neutrophils	3.85 ± 1.21	3.63 ± 1.01	3.55 ± 1.01	3.42 ± 1.05
Lymphocytes	2.26 ± 0.59	2.20 ± 0.56	2.16 ± 0.62	2.21 ± 0.59
Platelets	255.3 ± 50.9	258.7 ± 52.4	250.5 ± 50.0	252.7 ± 54.3
Monocytes	0.57 ± 0.16	0.64 ± 0.73	0.54 ± 0.33	0.56 ± 0.17
NLR	1.698 ± 0.490	1.715 ± 0.541	1.744 ± 0.608	1.619 ± 0.561
SII	435.7 ± 153.3	448.8 ± 182.09	435.9 ± 174.4	416.1 ± 197.4
PLR	117.4 ± 28.9	123.7 ± 34.9	124.1 ± 38.6	123.2 ± 47.9
MLR	0.260 ± 0.086	0.312 ± 0.434	0.262 ± 0.172	0.259 ± 0.088
	**Patients with BD (*n* = 321)**			
	**Spring (*n* = 90)**	**Summer (*n* = 51)**	**Autumn (*n* = 99)**	**Winter (*n* = 81)**
Sex	31/58	18/33	43/56	29/52
Age	49.4 ± 10.2	47.0 ± 10.7	46.6 ± 12.2	49.1 ± 11.4
BMI	26.58 ± 5.45	26.21 ± 4.62	24.69 ± 4.68	26.68 ± 4.99
Neutrophils	4.08 ± 1.25	4.27 ± 1.56	4.22 ± 1.42	4.34 ± 1.49
Lymphocytes	2.44 ± 0.80	2.45 ± 0.64	2.37 ± 0.83	2.40 ± 0.79
Platelets	235.7 ± 49.8	228.2 ± 64.2	243.4 ± 56.0	264.8 ± 73.07
Monocytes	0.64 ± 0.71	0.54 ± 0.17	0.62 ± 0.34	0.65 ± 0.63
NLR	1.801 ± 0.694	1.833 ± 0.748	2.145 ± 2.249	1.966 ± 0.908
SII	432.3 ± 219.2	426.6 ± 241.2	518.8 ± 433.3	531.7 ± 318.7
PLR	105.5 ± 40.9	99.2 ± 38.1	116.3 ± 55.3	122.1 ± 56.5
MLR	0.275 ± 0.239	0.229 ± 0.080	0.297 ± 0.257	0.276 ± 0.189
	**Patients with OCD (*n* = 255)**			
	**Spring (*n* = 67)**	**Summer (*n* = 61)**	**Autumn (*n* = 52)**	**Winter (*n* = 75)**
Sex	40/27	36/25	33/19	46/29
Age	36.1 ± 11.5	34.3 ± 12.5	36.7 ± 10.0	37.0 ± 11.5
BMI	25.21 ± 3.82	24.94 ± 5.38	24.56 ± 5.37	25.40 ± 4.50
Neutrophils	3.41 ± 1.04	3.42 ± 1.1	3.62 ± 1.27	3.39 ± 0.92
Lymphocytes	2.18 ± 0.69	2.28 ± 0.83	2.38 ± 0.68	2.35 ± 0.72
Platelets	235.01 ± 47.6	222.8 ± 36.9	237.7 ± 51.3	236.9 ± 55.8
Monocytes	0.50 ± 0.17	0.50 ± 0.16	0.57 ± 0.2	0.52 ± 0.14
NLR	1.687 ± 0.643	1.717 ± 1.09	1.582 ± 0.565	1.548 ± 0.582
SII	402.7 ± 192.9	381.3 ± 227.7	374.5 ± 148.5	365.8 ± 163.3
PLR	117.9 ± 46.5	109.3 ± 40.0	106.0 ± 30.8	108.6 ± 39.7
MLR	0.249 ± 0.090	0.236 ± 0.102	0.244 ± 0.068	0.236 ± 0.080

**Table 2 ijms-25-04310-t002:** Psychotropic treatments administered to the participants at the time of blood sampling, divided according to diagnosis and to season, and levels of significance of the observed differences.

**Effect of Diagnosis**								
	**Antidepressants**		**Antipsychotics**		**Lithium**		**Valproate**	
	**+**	**−**	**+**	**−**	**+**	**−**	**+**	**−**
BD (*n* = 321)	151 (47.04%)	170 (52.96%)	49 (15.26%)	272 (84.74%)	157 (48.91%)	164 (51.09%)	127 (39.56%)	194 (60.44%)
OCD (*n* = 255)	164 (64.31%)	91 (35.69%)	39 (15.29%)	216 (84.71%)	0 (0%)	255 (100%)	24 (9.41%)	231 (90.59%)
χ^2^; *p*	χ^2^ = 17.11; *p* < 0.0001		χ^2^ = 0.0009; *p* = 0.992		χ^2^ = 171.45; *p* < 0.0001		χ^2^ = 56.79; *p* < 0.0001	
**Effect of season—BD**								
Spring	44 (48.89%)	46 (51.1%)	14 (15.56%)	76 (84.44%)	44 (48.89%)	46 (51.11%)	36 (40%)	54 (60%)
Summer	21 (41.18%)	30 (58.82%)	6 (11.76%)	45 (88.24%)	25 (49.02%)	26 (50.98%)	23 (45.1%)	28 (54.9%)
Autumn	44 (44.44%)	55 (55.56%)	17 (17.17%)	82 (82.83%)	49 (49.49%)	50 (50.51%)	39 (39.39%)	60 (60.61%)
Winter	42 (51.85%)	39 (48.15%)	12 (14.81%)	69 (85.19%)	39 (48.15%)	42 (51.85%)	29 (35.80%)	52 (64.20%)
χ^2^; *p*	χ^2^ = 1.85; *p* = 0.605		χ^2^ = 0.78; *p* = 0.854		χ^2^ = 0.03; *p* = 0.998		χ^2^ = 1.14; *p* = 0.767	
**Effect of season—OCD**								
Spring	45 (67.16%)	22 (32.84%)	10 (14.93%)	57 (85.07%)	0	67	9 (13.43%)	58 (86.57%)
Summer	36 (59.02%)	25 (40.98%)	12 (19.67%)	49 (80.33%)	0	61	6 (9.84%)	55 (90.16%)
Autumn	33 (63.46%)	19 (36.54%)	9 (17.31%)	43 (82.69%)	0	52	3 (5.77%)	49 (94.23%)
Winter	50 (66.67%)	25 (33.33%)	8 (10.67%)	67 (89.33%)	0	75	6 (8.00%)	69 (92.00%)
χ^2^; *p*	χ^2^ = 1.18; *p* = 0.758		χ^2^ = 2.31; *p* = 0.510		NA		χ^2^ = 2.27; *p* = 0.519	

## Data Availability

The data presented in this study are available on request from the corresponding author. The data are not publicly available due to privacy and ethical restrictions.

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
