# Peer review of "Higher Seasonal Variation of Systemic Inflammation in Bipolar Disorder"

_ijms, 2024, doi:10.3390/ijms25084310_

Round 1
Reviewer 1 Report
Comments and Suggestions for Authors
In the manuscript of “Higher seasonal variation of systematic inflammation bipolar disorder,” the authors used the neurtrophil to lymphocyte ration (NLR) and the systemic inflammatory index (SII) to study the seasonal variations of systemic inflammation in patients with bipolar disorder, patients with obsessive-compulsive disorder (OCD), and healthy controls. The authors found that patients with bipolar disorder had significantly higher NLR and SSI in autumn and winter, but not in spring or summer compare to patients with OCD and healthy controls. The findings are interesting. However, the cross-sectional nature of each group and lack of clear information on the studied patients confounded the results. The manuscript was not well written so that it is difficult to understand what authors really wanted to address.
The authors seemingly confused the use of antidepressants in the treatment of bipolar disorder and major depressive disorder. The authors repeatedly stated the antidepressants in the treatment of bipolar disorder as well as major depressive disorder. It is true that “traditional” antidepressants can be use in the treatment of bipolar depression, but the authors should be very clear that the first-line medications for bipolar disorder are mood stabilizers (lithium, anticonvulsants, and antipsychotics), but not antidepressants.
Major depressive episode of bipolar disorder and major depressive disorder may share some underlying pathological changes including increase in inflammatory markers, but the response to “traditional” antidepressant is quite different. Therefore, the role of inflammatory markers in drug development and treatment response is likely different too. Grouping these two disorders as a mood disorder for future drug development is confusing.
As mentioned by the authors, the results could be confounded by the cross-sectional nature of the study. The discussion should focus on the validity of using this “new” method to study the inflammatory burden in patients with bipolar disorder instead of the possible mechanisms. There are a large number of studies of inflammatory changes in psychiatric disorders and seasonal variations in general population. The authors should discuss pros and cons of using NLR and SII to study the inflammatory changes in bipolar disorder as well as other psychiatric disorder compare with other commonly used measures such as CRP and cytokines.
The authors found a higher NLR, but not SSI in Spring and Summer in bipolar disorder than in patients with OCD. What does this mean? Both bipolar disorder and OCD were reported with increase in inflammatory burden. The authors need discuss these differences.
Seasonal variations of immune-inflammatory changes in healthy controls have been reported. Overall results suggest that neutrophils increase in winter and peak in January and low in summer, and lymphocytes increase spring and peak in March, and lower in autumn. In the current study, the NLR did not show this kind of change in healthy controls. Did the authors have any concerns about this finding?
The interference of comorbid anxiety in OCD to the NLR was reported (
NC, 2019). Patients with “pure” OCD are rare. More than 80% patients with OCD are likely to have major depressive disorder and/or other psychiatric disorders. How many patients were with other psychiatric disorders in the current study? What were the possible impact to the results? What medications patients were on at the blood collection?Similarly, more than 80% patients of bipolar disorder have other comorbid psychiatric disorders, mainly anxiety disorder. How many patients with bipolar disorder were with a psychiatric comorbidity? What were the possible impart to the results? In addition, how many patients with bipolar I, II, or other type at each season? What kind of mood states they were when they was admitted to the hospital? What mediations they were taking at the time of blood collection?
Was the study retrospective or prospective? The authors stated “We retrospectively studied….”at the beginning of the Materials and Methods, and then stated “After complete description of the study to the subjects, a written informed consent was obtained.” It is unclear how this could be happen.
In the Materials and Methods, the authors stated they calculated the effect size, but did not say how. In clinical trials, Cohen’s D is commonly used to calculate the effect size. The authors need provide a little more information on calculation of effect size in the current study.
The writing style is unconventional. There were so many long sentences across the text, which made it difficult to read and understand.
For the Table, the authors need add percentage (%) for M/F instead of using the number only.
The discussion should focus on the unmet need in this area instead of the hypothetical assumptions and speculations.
Comments on the Quality of English Languagetoo many long sentences.
Author Response
see file in attachment

Reviewer 2 Report
Comments and Suggestions for Authors
This topic is interesting but has not been thoroughly explored yet. However, there are some points that need improvement:
1) Figure 1 does not clearly indicate the represented value: mean? standard error or standard deviation?).
2)Please include information on the psychiatric medication administered to patients with BD and OCD, as well as any other possible comorbidities, in the demographic data.
3) The demographic data should include the body mass index. It would be interesting to know their contribution to the observed results and if possible, to discuss about it. In addition to having this information available to the reader.
4) This article (PMID: 33946871) could be useful to enrich the discussion.
Author Response
see file in attachment

Reviewer 3 Report
Comments and Suggestions for Authors
The article discusses the relationship between seasonal changes and immune-inflammatory markers in patients with Bipolar Disorder (BD). It highlights a pattern of severe illness episodes correlating with seasonal variations, with depressive and mixed episodes common in autumn and spring, and mania in summer. The study suggests a link between these patterns and changes in photoperiod affecting brain neurotransmitters. It also explores the role of immuno-inflammatory mechanisms in BD, showing that patients with BD have higher systemic inflammatory markers in autumn and winter. The study's findings point towards the potential of using immune-inflammatory biomarkers as tools for better understanding and treating BD. Study focuses on an innovative subject but needs some clarifications and revisions
1. The title of study should emphasize the fact that BD cases are all in depressive episode.
2. A significant concern in the study was using OCD patients as a control group under the assumption of their condition's non-seasonality. However, some literature reports suggest seasonal variations in OCD, contradicting this assumption. The study should reference sources confirming the lack of seasonality in OCD and critically evaluate the validity of using OCD patients as a control group.
3. The introduction implies a direct causal link between photoperiod changes and mood variations in Bipolar Disorder (BD). This assertion needs revision to acknowledge that the relationship may be correlational, not causal, and that other contributing factors could be involved.
4. The study's method, particularly regarding participant interviews, is unclear. Details such as the timing of interviews (at admission or discharge), the clinical status of patients (in remission or active phase), the interviewers' identity and their consensus, and protocols for excluding other psychiatric disorders need clarification.
5. Information on the power analysis for determining sample size is missing. The study should detail the number of participants approached, the reasons for exclusions, and how the final sample size was reached.
6. The criteria for participant inclusion and exclusion are unclearly articulated. These criteria, including the rationale behind the selection of certain drug groups for the OCD participants, should be more explicitly stated and explained.
7. The study should specify the exact months or periods when blood samples were collected for each season, as 'summer' and other seasonal terms can vary in interpretation.
8. The discussion section lacks a balanced view, particularly in addressing studies with conflicting results. It appears biased towards interpretations supporting the study's hypothesis. A more objective and critical discussion that considers all possible interpretations and conflicting evidence is necessary.
Author Response
see file in attachment

Round 2
Reviewer 1 Report
Comments and Suggestions for Authors
In the revised version, the authors have addressed some of my concerns, but there are still a number of issues needed to be addressed.
The authors provided a large amount of evidence supporting that they understood treatment of bipolar disorder including lithium, anticonvulsants, antipsychotics and antidepressants. Clearly, the authors missed my point. The authors argued that the discussion of these treatments is beyond the scope this study. The authors also argued that this study was not about treatment. However, it is unclear why the authors only mentioned antidepressants in the Introduction without mentioning other treatments as a justification for the study. If the authors used antidepressant use in bipolar disorder as an indication of treatment-resistance, the authors need provide more information on the definition of TRD in for bipolar disorders. Without a brief review of treatment of bipolar depression, using antidepressant alone in the Introduction can be misleading. At least, the authors should move some contents from the Discussion to the Introduction to emphasize why the antidepressants are important for their study.
The authors responded that all patients with OCD did not have a major depressive disorder and all patients with bipolar disorder did not have any psychiatric comorbidity. These homogenous presentations seemingly contradict previous studies worldwide and studies from Italy, i.e., psychiatric comorbidity in mood disorders and anxiety disorders is the role rather than the exception. Was patients with a psychiatric comorbidity excluded from the current study? If so, how many were excluded?
Per authors’ response, the study is likely a post-hoc analysis if they have publication(s) with the same group of patients or a prospective study if this is the first publication with this group of patients.
The authors argued that their writing style followed “standard guidelines about reporting scientifc results, ” but this statement may not be true. The following is just a few examples with lengthy sentences.
This sentence is from the Discussion with one sentence and one paragraph.
All these issues are open for further research, which may also address the main limitation of the present study, coming from its cross-sectional design, with one observation for each participant: prospective studies, assessing time-lagged variation of inflammatory markers across seasons in the same group with multiple sampling, are needed to clarify the specific patterns of change associated with these psychiatric conditions.
Following sentence is also from the Discussion.
Activation of innate immunity has a major impact on brain structure and function in BD, with levels of peripheral cytokines and immune cell composition associating with white matter microstructure, brain metabolites, effect of antidepressant treatments, and cognitive impairment [16, 57-59]; and also involving a decrease of 5-HT function mediated by tryptophan breakdown and by activation of the kynurenine pathway [7, 60, 61], which in turn impacts immune functions [62].
The following is also from the Discussion.
However, despite mood stabilizers being the mainstay for the treatment of depression in BD, antidepressant drugs are approved by the US Food and Drugs Administration (FDA) for the treatment of both unipolar or bipolar depression; and The British Association for Psychopharmacology [78], the French Society for Biological Psychiatry and Neuropsychopharmacology [79], the American Psychiatric Association [80], the Canadian Network for Mood and Anxiety Treatments (CANMAT) and the International Society for Bipolar Disorders [81], still consider the combination of antidepressants and lithium as possible option for this difficult condition, even if second-line in respect to mood stabilizers alone.
In addition
“….antidepressant drugs are approved by the US Food and Drugs Administration (FDA) for the treatment of both unipolar or bipolar depression…”
This statement is not true. No antidepressant monotherapy has been approved by the US FDA for bipolar depression. Only antidepressant approved for bipolar depression is the combination of olanzapine and fluoxetine.
Comments on the Quality of English LanguageToo many lengthy sentences which affect the quality of manuscript. It is also unconventional to use one sentence as a paragraph.
Author Response
Answer to Reviewer #1
Note to the Editors: The Reviewer did not articulate comments into separate issues. Single issues were extracted from the text, in order to clarify how we responded to them. In this process, we disregarded opinions, and focused on points that could be addressed.
- […] the authors need provide more information on the definition of TRD in for bipolar disorders. […]
We have already provided the requested information. We have already explicitly discussed and cited: (1) the ISBD/CANMAT guidelines for the treatment of bipolar depression; (2) the APA Practice Guidelines for the Treatment of Patients with Bipolar Disorder; (3) the French Society for Biological Psychiatry and Neuropsychopharmacology Task Force. Formal Consensus for the Treatment of Bipolar Disorder; (4) the Evidence-Based Guidelines for Treating Bipolar Disorder by the Consensus Group of the British Association for Psychopharmacology; and (5) the WFSBP Guidelines for the Biological Treatment of Bipolar Disorders. Moreover, we have also already cited the most recent Consensu on how to define TRD in clinical trails (Sforzini et al.), the product of a great effort to which we participated.
All the guidelines that we cited, coming from the most respected scientific societies in the world, discuss the issue of treatment resistance in bipolar disorder.
- […] Was patients with a psychiatric comorbidity excluded from the current study? If so, how many were excluded?
In the Methods section we have already explicitly listed absence of psychiatric comorbities among the inclusion criteria. Epidemiology of OCD, and rate of comorbid pathologies, are not among the scopes of the present papers, have not been studied here, and will not be discussed.
- […] The following is just a few examples with lengthy sentences. […] Too many lengthy sentences which affect the quality of manuscript. It is also unconventional to use one sentence as a paragraph
As per Reviewer’s request, we broke the criticized paragraphs into smaller sentences. We simply changed |;| with |.|
- In addition “….antidepressant drugs are approved by the US Food and Drugs Administration (FDA) for the treatment of both unipolar or bipolar depression…” This statement is not true. No antidepressant monotherapy has been approved by the US FDA for bipolar depression. Only antidepressant approved for bipolar depression is the combination of olanzapine and fluoxetine.
Given that this point (as we have already clarified) is of marginal interest in our paper, we decided to stop following the Reviewer interest in this discussion.
Therefore, the criticized statement was erased.
